# Potential Risk of Agrochemical Leaching in Areas of Edaphoclimatic Suitability for Coffee Cultivation

Gleissy Mary Amaral Dino Alves dos Santos [1], Antônio Augusto Neves [1], Maria Eliana Lopes Ribeiro de Queiroz [1], Vagner Tebaldi de Queiroz [2], Carlos Antonio Alvares Soares Ribeiro [1], Efraim Lázaro Reis [1], Ana Carolina Pereira Paiva [1], José Romário de Carvalho [1], Samuel Ferreira da Silva [2], Ronie Silva Juvanhol [3], Taís Rizzo Moreira [2], Luciano José Quintão Teixeira [2], Sérgio Henriques Saraiva [2], Adilson Vidal Costa [2], Camila Aparecida da Silva Martins [2], Fábio Ribeiro Pires [2], Thuelem Azevedo Curty [2], Plinio Antonio Guerra Filho [4,*], Marcelo Henrique de Souza [5], Waldir Cintra de Jesus Junior [6] and Alexandre Rosa dos Santos [2]

[1] Chemistry Department, Viçosa Campus, Federal University of Viçosa/UFV, Viçosa 36570-000, MG, Brazil; gleissym@yahoo.com.br (G.M.A.D.A.d.S.); aneves@ufv.br (A.A.N.); meliana@ufv.br (M.E.L.R.d.Q.); cribeiro@ufv.br (C.A.A.S.R.); efraimreis@gmail.com (E.L.R.); anacarolinapaiva2@gmail.com (A.C.P.P.); jromario_carvalho@hotmail.com (J.R.d.C.)

[2] Rural Engineering Department, Alegre Campus, Federal University of Espírito Santo/UFES, Alegre 29500-000, ES, Brazil; wagnertq@gmail.com (V.T.d.Q.); samuelfd.silva@yahoo.com.br (S.F.d.S.); taisr.moreira@hotmail.com (T.R.M.); luqteixeira@yahoo.com.br (L.J.Q.T.); sergio.saraiva@ufes.br (S.H.S.); avcosta@hotmail.com (A.V.C.); camila.cca@hotmail.com (C.A.d.S.M.); fabio.pires@ufes.br (F.R.P.); thuelem.curty@hotmail.com (T.A.C.); alexandre.santos@pq.cnpq.br (A.R.d.S.)

[3] Forest Engineering Department, Professora Cinobelina Elvas Campus, Federal University of Piauí/UFPI, Bom Jesus 64900-000, PI, Brazil; roniejuvanhol@gmail.com

[4] Agricultural Engineering Department, Agricultural and Environmental Sciences Center, Federal University of Maranhão/UFMA, Chapadinha 65500-000, MA, Brazil

[5] Secretariat of Education of the State of Espírito Santo/SEDU, Guaçuí 29560-000, ES, Brazil; marcelosouza.rec@hotmail.com

[6] Agronomic Engineering Department, Center for Natural Sciences, Federal University of São Carlos/UFSCar, Campus Lagoa do Sino, Aracaçu, Buri 18290-000, SP, Brazil; wcintra@pq.cnpq.br

\* Correspondence: plinioguerraf@hotmail.com; Tel.: +55-19-981030740

**Abstract:** Studies show that agricultural activities around the world still present a strong dependence on agrochemicals that can leach into the soil profile, causing its contamination, as well as that of water resources. In this context, the present study evaluates the potential risk of pesticide leaching in areas of edaphoclimatic suitability for coffee cultivation in Espírito Santo state, Brazil. As a methodology, the areas of edaphoclimatic suitability for conilon and arabica coffee were defined, and subsequently, the risk of leaching of active agrochemical ingredients in these areas was evaluated using the Groundwater Ubiquity Score (*GUS*), Leaching Index (*LIX*) and Attenuation Factor/Retardation Factor (*AF/RF*) methods. Of the ten active ingredients evaluated, sulfentrazone and thiamethoxam present a potential risk of leaching into the groundwater level. The study allowed us to evaluate the potential risk of agrochemical leaching in tropical soils cultivated with coffee using geographic information system (*GIS*) techniques. The methodological proposal can be adapted for other agricultural areas and crops.

**Keywords:** pesticides; environmental pollution; water pollution; soil pollution; GIS

## 1. Introduction

The use of chemicals for pest control has intensified in recent years, mainly due to increased production and greater profit generation by farmers [1]. As a consequence, agriculture has become one of the major soil and aquatic system contaminants due to the high amount of agrochemicals used [2]. Previous analytical research has shown water

contamination by pesticides [3–5] and recent studies found in the literature indicate an increase in the evidence of contamination of water resources and soils by pesticides and some of their metabolites [6–11].

The leaching of pesticides into the soil is directly related to the physicochemical characteristics of the soil and to those of the agrochemical itself [7,12,13]. It is known that chemical compounds with basic characteristics are more easily fixed in soil than those with acidic characteristics [12,14–17]. Relationships between solubility and sorption of pesticides and their fixation in the soil are also relevant [15,18]. Physicochemical soil characteristics, such as the presence of minerals and clay, organic matter, pH, oxidative characteristics and general soil composition, may influence the retention or breakdown of agrochemicals [14,15,17,19–21]. The half-life of the agrochemical in the soil evaluates the time required for half the agrochemical concentration to disappear, regardless of its initial concentration in the environment [22].

Soil management methods for crop irrigation are among the factors that lead to the increase in contamination of groundwater by leaching [23]. Therefore, human action is an aggravating factor of contamination [6,24,25]. In order to avoid such risks, the use of pesticides should be reduced or eliminated altogether, especially in sites close to water collection areas [2].

Another way of minimizing contamination of soils and waters by pesticides is to minimize the loss of these products through leaching [2]. In order to evaluate the risk of leaching of pesticides, several models may be employed, such as *GUS* (Groundwater Ubiquity Score) [26] and *LIX* (Leaching Index) [27], which are simpler models based on physicochemical characteristics of the pesticide in question, or *AF/RF* (Attenuation Factor/Retardation Factor) [28], which is a more complex model based on the physicochemical characteristics of the pesticide studied, soil characteristics and geoclimatic conditions of the study area [29].

For these models, some coefficients are used, from an environmental point of view, the most used partition coefficients are the air-water partition coefficients ($K_H$ or $K_{aw}$) and the octanol-water partition coefficients ($K_{ow}$). These are used to describe the transport of chemicals between water and organic phases, such as lipids and natural organic carbon. Other partition coefficients that are also widely used include octanol-air, sediment-water, organic material-water, lipid-water, aerosol-air, and soil-water partitions [30]. According to [31], one of the reasons why alkane/water partition coefficients are used in environmental studies is their ability to mimic the hydrophilic and hydrophobic limits found in nature. Another important tool in estimating the sorption potential of dissolved contaminant in contact with the soil is the partition coefficient ($K_d$), and the higher the $K_d$, greater the tendency of contaminant to be adsorbed to the soil or sediment [32].

In recent years, Geographic Information Systems (*GIS*) techniques have helped to obtain information on pesticide leaching through a database containing georeferenced information for the evaluation of water resources, with a focus on groundwater [33–36]. The combination of multidisciplinary techniques has proven to be useful to minimize problems of spatial and temporal variation in parameters involved in the leaching of agrochemicals into underground aquatic resources [23,37].

With the aim of supporting decision-making for diagnosis, planning and governmental management, studies of areas presenting the potential risk of pesticide leaching become essential for the strategic establishment of mitigation and integrated pest, disease and weed management programs. In this context, the present study allows us to evaluate the potential risk of pesticide leaching in areas of edaphoclimatic suitability for coffee cultivation in the state of Espírito Santo, Brazil, with an emphasis on the most used pesticides, which may present leaching potential at the groundwater level.

## 2. Materials and Methods

### 2.1. Study Area

The study area is represented by the Espírito Santo state, Brazil, which has a territorial area of 46,053.19 km$^2$. It is located between the parallels of 17°53′29″ and 21°18′03″ South latitude and the meridians 39°41′18″ and 41°52′45″ West of Greenwich (Figure S1, Supplementary Material). Along with the states of Minas Gerais, Rio de Janeiro and São Paulo, it makes up the so-called Southeast Region Development Band [38].

The state has four climatic types according to the Köppen classification: (a) Cwb: subtropical climate, with dry winters and mild summers found in the mountainous region of the state; (b) Cwa: subtropical climate with dry winters and hot summers found in the southwest region of the state; (c) Am: humid or subtropical humid climate found in the northeast region of the state; and (d) Aw: tropical climate, with dry winters found in the western region of the state [39].

### 2.2. Edaphoclimatic Zoning for Coffee in Espírito Santo State, Brazil

The edaphoclimatic zoning for conilon and arabica coffee cultivation was carried out according to the methodology proposed by [40] using information collected in specialized literature [41–47] (Tables S1–S3, Supplementary Material).

The methodological flowchart containing the seven sub steps required for the establishment of the edaphoclimatic zoning for conilon and arabica coffee in Espírito Santo state is presented in Figure S2 of Supplementary Material.

### 2.3. Evaluation of Potential Leaching Risk Using the LIX and GUS Methods

The objective of this study was to evaluate the risk of agrochemical leaching in areas of soil and climate suitability for conilon and arabica coffee cultivation in Espírito Santo state, Brazil. The evaluated pesticides contained active ingredients authorized by the National Health Vigilance Agency of Brazil (ANVISA), supervised by the Ministry of Agriculture, Livestock and Supply of Brazil (MAPA) and the Institute of Agricultural and Forest Defense (IDAF) of the Espírito Santo state, Brazil, with an emphasis on: (a) herbicides—2,4 D, diuron, glyphosate, paraquat, pendimethalin and sulfentrazone; (b) insecticides—chlorpyrifos, terbufos and thiamethoxam and; (c) fungicide—tebuconazole.

In this step, the *GUS* and *LIX* methods were used to evaluate the leaching potential of these 10 active ingredients of agrochemicals used in coffee production. Therefore, the *GUS* method, proposed by [26], is expressed by Equation (1):

$$GUS = \log\left[\left(t\frac{1}{2}soil\right) \times (4 - \log K_{OC})\right] \tag{1}$$

where, $t\frac{1}{2}soil$ is the half-life of the product in the soil, in days$^{-1}$; and $K_{OC}$ is the coefficient of adsorption to organic carbon, in mL·g$^{-1}$.

Subsequently, the *GUS* value obtained for each active ingredient was classified into one of the categories defined by pre-established ranges, according to the following intervals:

(a)  $GUS \leq 1.8$ = does not undergo leaching;
(b)  $1.8 < GUS < 2.8$ = transition range;
(c)  $GUS \geq 2.8$ = probable leaching.

The *LIX* method, proposed by [27], is expressed by the Equation (2):

$$LIX = exp(-k \times K_{OC}) \tag{2}$$

where $k$ is the first order breakdown constant of the pesticide in the soil, in days$^{-1}$ (Equation (3)); and $K_{OC}$ is the coefficient of adsorption to organic carbon, in mL·g$^{-1}$.

The first-order breakdown constant of the pesticide in the soil is expressed by Equation (3):

$$k = \frac{\ln 2}{t_{\frac{1}{2}}} \tag{3}$$

The value of *LIX* obtained for each active ingredient was classified according to the intervals:

(a)  0 = null;
(b)  0 to 0.1 = transition zone;
(c)  ≥ 0.1 = leaching potential.

The physicochemical properties of the 10 active pesticides used in the study for coffee cultivation in Espírito Santo state using the *GUS* and *LIX* methods are presented in Table 1.

**Table 1.** Physicochemical properties of active ingredients used in coffee cultivation.

| Pesticides | Soil $t_{\frac{1}{2}}$ (days$^{-1}$) | $K_{OC}$ (mL g$^{-1}$) | $k$ (days$^{-1}$) | $K_H$ (Pa m$^3$ mol$^{-1}$) |
|---|---|---|---|---|
| 2,4-D [48] | 4.4 | 39.3 | 0.157533 | $4.0 \times 10^{-6}$ |
| Chlorpyrifos [48] | 50.0 | 8151.0 | 0.013863 | $4.78 \times 10^{-1}$ |
| Diuron [48] | 75.5 | 813,0 | 0.009181 | $2.00 \times 10^{-6}$ |
| Glyphosate [48] | 15.0 | 1424.0 | 0.046210 | $2.10 \times 10^{-7}$ |
| Paraquat [48] | 3000.0 | 1,000,000.0 | 0.001899 | $4.0 \times 10^{-9}$ |
| Pendimethalin [48] | 182.3 | 17,491.0 | 0.003802 | $2.73 \times 10^{-3}$ |
| Sulfentrazone [48,49] | 541.0 | 43.0 | 0.001281 | $1.878 \times 10^{-4}$ |
| Tebuconazole [48,50] | 63.0 | 769.0 | 0.011002 | $1.00 \times 10^{-5}$ |
| Terbufos [48] | 8.0 | 500.0 | 0.086643 | 2.70 |
| Thiamethoxam [48] | 50.0 | 56.2 | 0.013863 | $4.70 \times 10^{-10}$ |

### 2.4. Specialization and Evaluation of the Risk of Leaching Using RF/AF

In step 1, the spatial distribution of the replenishment rate of surface water was performed. In this stage, in the ArcGIS® software (www.esri.com (accessed date: 15 September 2016)), the meteorological variables Potential Evapotranspiration of Culture (ETC) and Rainfall (P), originating from the agroclimatic water balance proposed by [51] were initially converted from the tabular format into the vector point format using the "add X, Y data" function. Subsequently, the "spatial interpolation by the Inverse of the Distance Square (IQD)" function was applied to generate the ETC and P matrix images. For the water replenishment (q) matrix image, the P matrix image was subtracted from the ETC image using the "map algebra" function.

In Step 2, the spatial distribution of the Retardation Factor (*RF*) for pesticide movement in the soil was performed. On the polygonal vector map of soil types of the Espírito Santo state, in its attribute table, the representative fields of physicochemical properties of the active ingredients (Table 1) and soil physicochemical variables (Table S1, Supplementary Material) for the calculation of the *RF* model according to Equation (4):

$$RF = 1 + \left( \frac{\rho \times OC \times K_{OC}}{FC} \right) + \left( \frac{\delta \times K_H}{FC} \right) \tag{4}$$

where $\rho$ is the soil density, in g·cm$^{-3}$; *OC* is the organic carbon content, in g·g$^{-1}$; $K_{OC}$ is the coefficient of adsorption to organic carbon, in cm$^3$·g$^{-1}$; *FC* is the field capacity, in v·v$^{-1}$; $\delta$ is the soil porosity in field capacity, in v·v$^{-1}$; and $K_H$ is the air-water partition coefficient of the pesticide, in cm$^3$·g$^{-1}$.

After the addition of these variables to the model, the "field calculator" function was used for spatial *RF* processing of the 10 active ingredients evaluated.

The polygonal *RF* vector images were rasterized ("polygons to raster" function) and reclassified ("reclassify" function) according to the adsorption potential's classes (Table S4, Supplementary Material).

Finally, the *RF* maps in the polygonal vector format were cut ("cut" function) in relation to the edaphoclimatic suitability areas generating the final *RF* maps for the edaphoclimatic suitability areas for conilon and arabica coffee in Espírito Santo state.

In Step 3, spatial distribution of soil-based pesticide attenuation (*AF*) was performed. In this stage, the representative image of the HAND (Height Above the Nearest Drainage) model using the topographic information of the Digital Elevation Model (DEM) from Shuttle Radar Topography Mission (SRTM), water flow direction, water flow accumulation and drainage was obtained to extract hydrologically consistent information from an area necessary to identify the groundwater depth from the surface (*L*). It was also considered for the spatialization of *AF*, with the effective root system depth of conilon and arabica coffee being assumed as 0.3 m.

The water replenishment (*q*) matrix image, considering annual irrigation (I) of 600 and 1200 mm, and the *RF*, *FC*, $t_{\frac{1}{2}}soil$ and *L* matrix images were added as independent variables, using the "map algebra" function in the *AF* model for the 10 active ingredients evaluated according to Equation (5):

$$AF = exp\left( \frac{-0.693 \times L \times RF \times FC}{q \times t_{\frac{1}{2}}soil} \right) \qquad (5)$$

where *L* is the groundwater depth (or depth considered) from the ground surface, in m; *RF* is the Retardation Factor, dimensionless; *FC* is the soil field capacity, in v·v$^{-1}$; *q* is the net replenishment of groundwater, in m·day$^{-1}$; and $t_{\frac{1}{2}}$ *soil* is the half-life of the product in the soil, in days.

The *AF* matrix images were reclassified ("reclassify" function) according to the leaching potential's classes (Table S5, Supplementary Material).

Finally, the *AF* maps in the polygonal vector format were cut ("cut" function) in relation to the edaphoclimatic suitability areas generating the final *AF* maps of the edaphoclimatic suitability areas for conilon and arabica coffee cultivation in Espírito Santo state.

Figure S3 of Supplementary Material presents the methodological steps necessary for spatialization and the evaluation of leaching risk of agrochemicals active ingredients using the *RF/AF* method in the edaphoclimatically suitable areas cultivated with conilon and arabica coffee in Espírito Santo state.

*2.5. Evaluation of Leaching Risk for the Main Coffee-Producing Municipalities*

The spatialization and the evaluation of leaching risk of agrochemicals active ingredients was realized for areas cultivated with conilon and arabica coffee in the main producing municipalities in Espírito Santo state, Brazil using the *RF/AF* method. The study was developed in two stages, both aided by the use of ArcGIS® software, version 10.4, ArcMap module. In the first stage, the areas planted with conilon coffee were mapped in the municipalities of Jaguaré, Vila Valério and Sooretama, considered to be the largest conilon coffee producers in Espírito Santo state (Figure S4, Supplementary Material). This same procedure was performed for the municipalities of Brejetuba, Ibatiba and Iúna, considered the largest arabica coffee producers (Figure S5, Supplementary Material). In the second stage, the leaching risk of active ingredients of agrochemicals in the areas mapped that presented coffee cultivation was evaluated using the *RF/AF* method.

The mapping of areas cultivated with coffee was obtained by photointerpretation [52] on the orthophotomosaic of 2007, provided by the State Institute of Environment and Water Resources (IEMA), with a spatial resolution of 1 m. The cartographic scale used was 1:1500.

The cartographic projections and the original data of this database was transformed into Mercator Transverse Universal Mapping (UTM) and Horizontal Datum SIRGAS 2000, to comply with Decree No. 5334/2005 and Resolution No. 1/2005 of the Brazilian Institute of Geography and Statistics (IBGE), which establish SIRGAS 2000 as the new Geocentric Reference System for Brazil [53].

The second step consisted of the delimitation ("mask extraction" function) of the leaching risk images of the active ingredients sulfentrazone and thiamethoxam for the photointerpreted areas with conilon and arabica coffee obtained in the first step.

Figure S6 of Supplementary Material presents the methodological flowchart containing the necessary steps for spatialization and the evaluation of leaching risk of agrochemicals active ingredients using the *RF/AF* method in areas cultivated with coffee in the main production municipalities in Espírito Santo state.

### 2.6. Multivariate Correspondence Analysis

The percentage area data for the *AF* indices of the active ingredients sulfentrazone and thiamethoxam for each coffee species were analyzed by means of multivariate correspondence analysis considering soil type and scenario (depth and irrigation). The same analysis was used for each active ingredient in the representative municipalities for each coffee species, based on the proposed scenarios.

Analyses were performed on vegan packages [54] of the R application, version 3.4 [55].

## 3. Results

### 3.1. Edaphoclimatic Zoning for Coffee

The preliminary mapping—required for elaboration of edaphoclimatic zoning for conilon and arabica coffee in Espírito Santo state, Brazil—took into account the edaphoclimatic variables for the state and the edaphoclimatic suitability ranges for conilon and arabica coffees, which are presented in Figures S7–S9 of Supplementary Material, respectively. Figures S10 and S11 of Supplementary Material presents the edaphoclimatic zoning for conilon and arabica coffee in the study area.

### 3.2. Evaluation Using the LIX and GUS Methods

Table 2 presents the potential risk of leaching of the active ingredients considered in the study by the *LIX* and *GUS* methods. The active ingredients sulfentrazone and thiamethoxam presenting potential and probable leaching in the soil were highlighted by both methods.

**Table 2.** Potential risk of agrochemical leaching into soil evaluated by the LIX and GUS methods.

| Pesticides | LIX | Classification * | GUS | Classification ** |
|---|---|---|---|---|
| 2,4-D | 0.00000 | N | 1.55 | NL |
| Chlorpyrifos | 0.00000 | N | 0.15 | NL |
| Diuron | 0.00057 | ZT | 2.05 | FT |
| Glyphosate | 0.00000 | N | 1.00 | NL |
| Paraquat | 0.00000 | N | −6.95 | NL |
| Pendimethalin | 0.00000 | N | −0.55 | NL |
| Sulfentrazone | 0.94640 | PL | 6.47 | NL |
| Tebuconazole | 0.00021 | ZT | 2.00 | FT |
| Terbufos | 0.00000 | N | 1.17 | NL |
| Thiamethoxam | 0.45882 | PL | 3.82 | PL |

* N: Null, ZT: Transition Zone, PL: Leaching Potential. ** NL: No Leaching, FT: Transition Range, PL: Probable leaching.

### 3.3. Spatialization and Evaluation Using the RF/AF Method

By the *LIX* and *GUS* methods the active ingredients sulfentrazone and thiamethoxam presented leaching potential. Therefore, these products were analyzed by *RF* and presented low adsorption potential for suitable areas of the edaphoclimatic zoning for conilon and arabica coffee (Figures S12 and S13, Supplementary Material).

The Attenuation Factor (*AF*) indices for estimation of the Potential Risk of Leaching (PRL) of the ten active agrochemical ingredients evaluated for conilon and arabica coffee production are presented in Figures 1 and 2, indicating representative values for the classes ranging from very low to very high PRL.

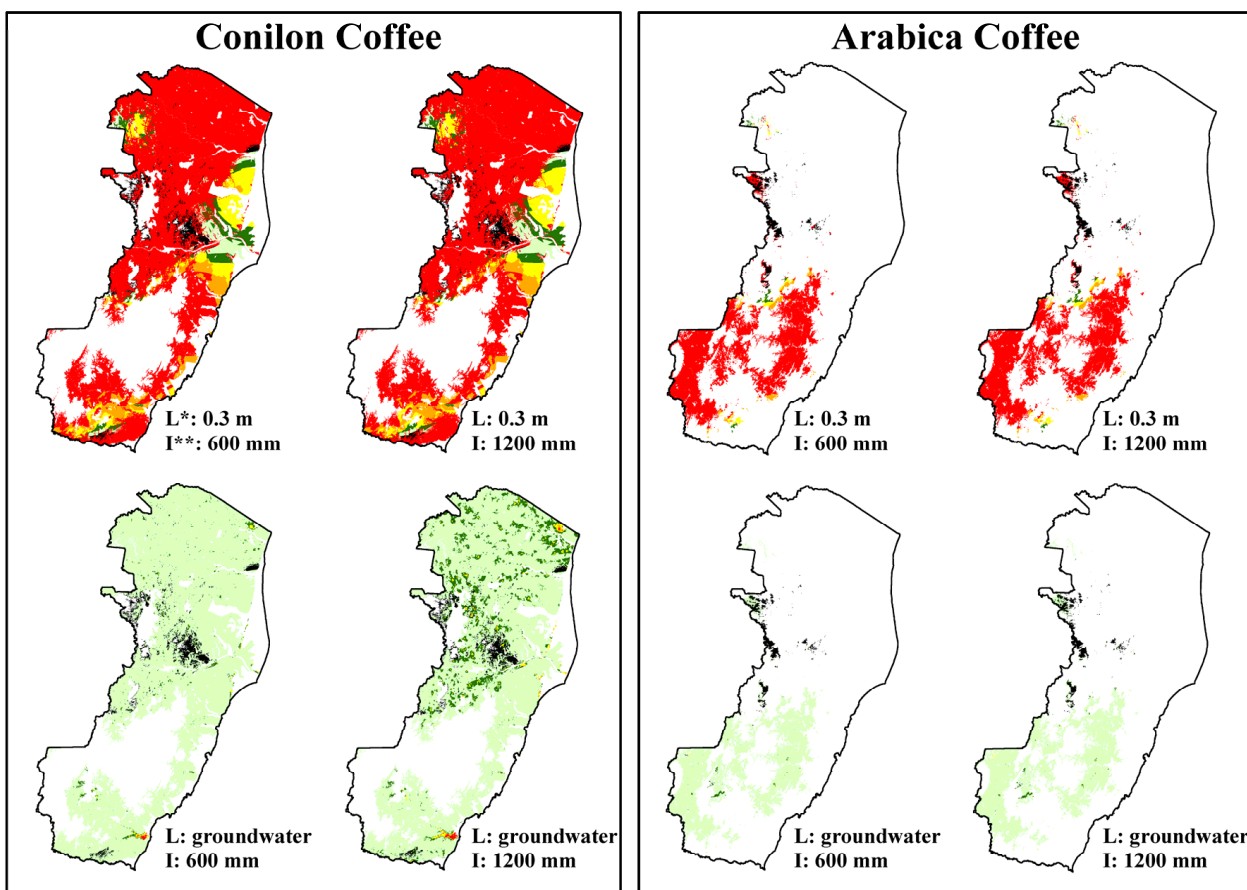

| Depth (m) | Legend | Potential leaching Sulfentrazone | Attenuation Factor (AF) | Conilon Coffee | | | | Arabica Coffee | | | |
|---|---|---|---|---|---|---|---|---|---|---|---|
| | | | | 600 mm | | 1200 mm | | 600 mm | | 1200 mm | |
| | | | | Area (km²) | % | Area (km²) | % | Area (km²) | % | Area (km²) | % |
| 0.3 | | Very low | 0.0 to 0.0001 | 994.66 | 3.30 | 994.66 | 3.30 | 28.25 | 0.36 | 28.25 | 0.36 |
| | | Low | 0.0001 to 0.01 | 1330.18 | 4.41 | 1330.18 | 4.41 | 100.12 | 1.26 | 100.12 | 1.26 |
| | | Medium | 0.01 to 0.1 | 2128.09 | 7.06 | 2128.09 | 7.06 | 141.77 | 1.79 | 141.77 | 1.79 |
| | | High | 0.01 to 0.25 | 1816.04 | 6.03 | 1816.04 | 6.03 | 291.51 | 3.67 | 291.51 | 3.67 |
| | | Very high | 0.25 to 1 | 22,769.45 | 75.55 | 22,769.45 | 75.55 | 6917.63 | 87.18 | 6917.63 | 87.18 |
| | | Null and Improper areas | - | 1101.07 | 3.65 | 1101.07 | 3.65 | 455.33 | 5.74 | 455.33 | 5.74 |
| | | **Total** | | **30,139.49** | **100.00** | **30,139.49** | **100.00** | **7934.61** | **100.00** | **7934.61** | **100.00** |
| Groundwater level | | Very low | 0.0 to 0.0001 | 28,502.84 | 94.57 | 25,617.07 | 85.00 | 7413.14 | 93.43 | 7383.18 | 93.05 |
| | | Low | 0.0001 to 0.01 | 407.43 | 1.35 | 2916.63 | 9.68 | 61.28 | 0.77 | 86.78 | 1.09 |
| | | Medium | 0.01 to 0.1 | 88.55 | 0.29 | 381.89 | 1.27 | 4.51 | 0.06 | 8.72 | 0.11 |
| | | High | 0.01 to 0.25 | 20.56 | 0.07 | 69.68 | 0.23 | 0.30 | 0.004 | 0.53 | 0.01 |
| | | Very high | 0.25 to 1 | 19.04 | 0.06 | 53.14 | 0.18 | 0.05 | 0.001 | 0.06 | 0.001 |
| | | Null and Improper areas | - | 1101.07 | 3.65 | 1101.07 | 3.65 | 455.33 | 5.74 | 455.33 | 5.74 |
| | | **Total** | | **30,139.49** | **100.00** | **30,139.49** | **100.00** | **7934.61** | **100.00** | **7934.61** | **100.00** |

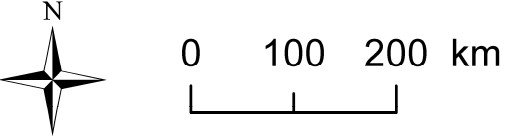

Universal Transverse Mercator Projection
Ellipsoid: SIRGAS 2000
ZONE 24S

**Figure 1.** Attenuation Factor (*AF*) of active ingredient sulfentrazone evaluated for edaphoclimatically suitable areas for conilon (*Coffea canephora* Pierre ex Froehner) and arabica (*Coffea arabica* L.) coffee in Espírito Santo state, Brazil. * L: Groundwater depth and ** I: Irrigation.

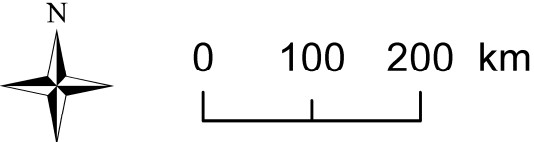

| Depth (m) | Legend | Potential leaching Thiamethoxam | Attenuation Factor (AF) | Conilon Coffee | | | | Arabic Coffee | | | |
|---|---|---|---|---|---|---|---|---|---|---|---|
| | | | | 600 mm | | 1200 mm | | 600 mm | | 1200 mm | |
| | | | | Area (km²) | % | Area (km²) | % | Area (km²) | % | Area (km²) | % |
| 0.3 | | Very low | 0.0 to 0.0001 | 9843.70 | 32.66 | 9651.50 | 32.02 | 3868.39 | 48.75 | 3859.50 | 48.64 |
| | | Low | 0.0001 to 0.01 | 11,672.37 | 38.73 | 1099.76 | 3.65 | 3176.97 | 40.04 | 2980.23 | 37.56 |
| | | Medium | 0.01 to 0.1 | 7190.22 | 23.86 | 7308.36 | 24.25 | 175.07 | 2.21 | 181.65 | 2.29 |
| | | High | 0.01 to 0.25 | 323.64 | 1.07 | 10,870.46 | 36.07 | 253.98 | 3.20 | 453.04 | 5.71 |
| | | Very high | 0.25 to 1 | 8.49 | 0.03 | 108.34 | 0.36 | 4.86 | 0.06 | 4.86 | 0.06 |
| | | Null and Improper areas | - | 1101.07 | 3.65 | 1101.07 | 3.65 | 455.33 | 5.74 | 455.33 | 5.74 |
| | | **Total** | | **30,139.49** | **100.00** | **30,139.49** | **100.00** | **7934.61** | **100.00** | **7934.61** | **100.00** |
| Groundwater level | | Very low | 0.0 to 0.0001 | 29,033.47 | 96.33 | 29,019.92 | 96.29 | 7479.28 | 94.26 | 7479.28 | 94.26 |
| | | Low | 0.0001 to 0.01 | 4.84 | 0.02 | 13.85 | 0.05 | - | - | - | - |
| | | Medium | 0.01 to 0.1 | 0.11 | 0.00 | 4.57 | 0.02 | - | - | - | - |
| | | High | 0.01 to 0.25 | - | - | 0.07 | 0.00 | - | - | - | - |
| | | Very high | 0.25 to 1 | - | - | - | - | - | - | - | - |
| | | Null and Improper areas | - | 1101.07 | 3.65 | 1101.07 | 3.65 | 455.33 | 5.74 | 455.33 | 5.74 |
| | | **Total** | | **30,139.49** | **100.00** | **30,139.49** | **100.00** | **7934.61** | **100.00** | **7934.61** | **100.00** |

N

0    100   200 km

Universal Transverse Mercator Projection
Ellipsoid: SIRGAS 2000
ZONE 24S

**Figure 2.** Attenuation Factor (*AF*) of active ingredient thiamethoxam evaluated for edaphoclimatically suitable areas for conilon (*Coffea canephora* Pierre ex Froehner) and arabica (*Coffea arabica* L.) coffee in Espírito Santo state, Brazil. * L: Groundwater depth and ** I: Irrigation.

Among the ten active ingredients evaluated, sulfentrazone presented the largest area for the very high PRL to a depth of 0.30 m in both irrigations (600 and 1200 mm), both for conilon (75.55% $\approx$ 22.769.45 km$^2$) and arabica (87.18% $\approx$ 6917.63 km$^2$) coffee (Figure 1). For conilon coffee, considering groundwater depth, values equivalent to 0.06% (19.04 km$^2$) and zero, were found for the very high PRL class, when simulating irrigations of 600 and 1200 mm, 18% (53.14 km$^2$), respectively. However, for arabica coffee, the results found for the very high PRL class were lower, presenting approximately 0.001% for both irrigation depths.

For the active ingredient thiamethoxam (Figure 2) the results obtained also revealed values for all PRL classes for the depth of 0.30 m. Among the active ingredients evaluated, thiamethoxam had the highest distribution for the PRL classes, standing out with the irrigation depth of 1200 mm with 36.07% (10,8670.46 km$^2$) of the high PRL class for conilon coffee. For arabica coffee, the groundwater and irrigation levels (600 and 1200 mm) meant that the potential for leaching remained low in relation to the other classes. For conilon coffee, when simulating an irrigation depth of 1200 mm, 0.02% (4.57 km$^2$) of the area presented medium PRL.

Figure S14 of Supplementary Material presents the relation of the *AF* index with the soil types for the main active ingredients with PRL. In relation to conilon coffee, for the root system depth of 0.30 m and irrigation of 600 mm, all soil types presented limits above 0.25 (very high PRL class), except for the haplic organosol and cambisol soil types. For irrigation of 1200 mm, all soil types presented very high PRL. It is also important to highlight that thiamethoxam has a high PRL for the following conditions: root system depth of 0.30 m and irrigation depth of 600 mm, for litholic neosols; and for irrigation depth of 1200 mm, for fluvic neosoil and litholic neosoil. When considering the groundwater depth for both irrigations, only fluvic neosol soil stands out as high PRL (0.1 to 0.25).

For arabica coffee at a root depth of 0.30 m in both irrigations (Figure S14, Supplementary Material), all soil types presented limits in the very high PRL class for sulfentrazone. For thiamethoxam, litholic neosol soil stands out in the high PRL class. Considering the groundwater depth for both irrigations, all soil types presented low or very low PRL.

The multivariate correlation analysis of the percentage area of the *AF* index for the active ingredients sulfentrazone and thiamethoxam taking into account soil types, proposed scenarios (depth and irrigation) and conilon or arabica coffee cultivars, demonstrated independence ($p > 0.05$) (Figure S15, Supplementary Material).

### 3.4. Assessment of Leaching Risks for the Main Conilon Coffee Producing Municipalities

The *AF* indexes of the sulfentrazone active ingredient evaluated for areas cultivated with conilon coffee for the municipalities of Jaguaré, Vila Valério and Sooretama, ES, are presented in Figures 3 and 4. In the municipality of Vila Valério, in areas cultivated with conilon coffee with a root system depth of 0.30 m, and simulating 600 and 1200 mm depth irrigations, a very high PRL class equivalent to 99.24% (112.45 km$^2$) was observed, followed by the municipalities of Jaguaré and Sooretama (59.71% = 72.94 km$^2$ and 20.88% = 15.94 km$^2$, respectively). For groundwater level depth, for both irrigations, none of the three municipalities presented areas with very high PRL. However, it was observed that the municipalities of Vila Valério and Jaguaré presented areas with high (0.01% = 0.02 km$^2$) and average (1.11% = 1.36 km$^2$) potential risk of leaching, respectively, for irrigation of 1200 mm. Figure 5 and Figure S16 of Supplementary Material show the evaluation of the *AF* index for the active ingredient thiamethoxam evaluated for areas cultivated with conilon coffee in the municipalities of Jaguaré, Vila Valério and Sooretama. For areas cultivated with conilon coffee at a root system depth of 0.30 m, simulating irrigations of 600 and 1200 mm, the municipality of Vila Valério presented the very high PRL class (5.36% = 6.07 km$^2$). However, for the 1200 mm irrigation, the municipalities of Jaguaré and Sooretama presented areas with a lower PRL (3.36% = 4.11 km$^2$ and 0.30% = 0.23 km$^2$, respectively). For the groundwater depth, for both irrigations (600 and 1200 mm), the three municipalities only presented areas with very low PRL.

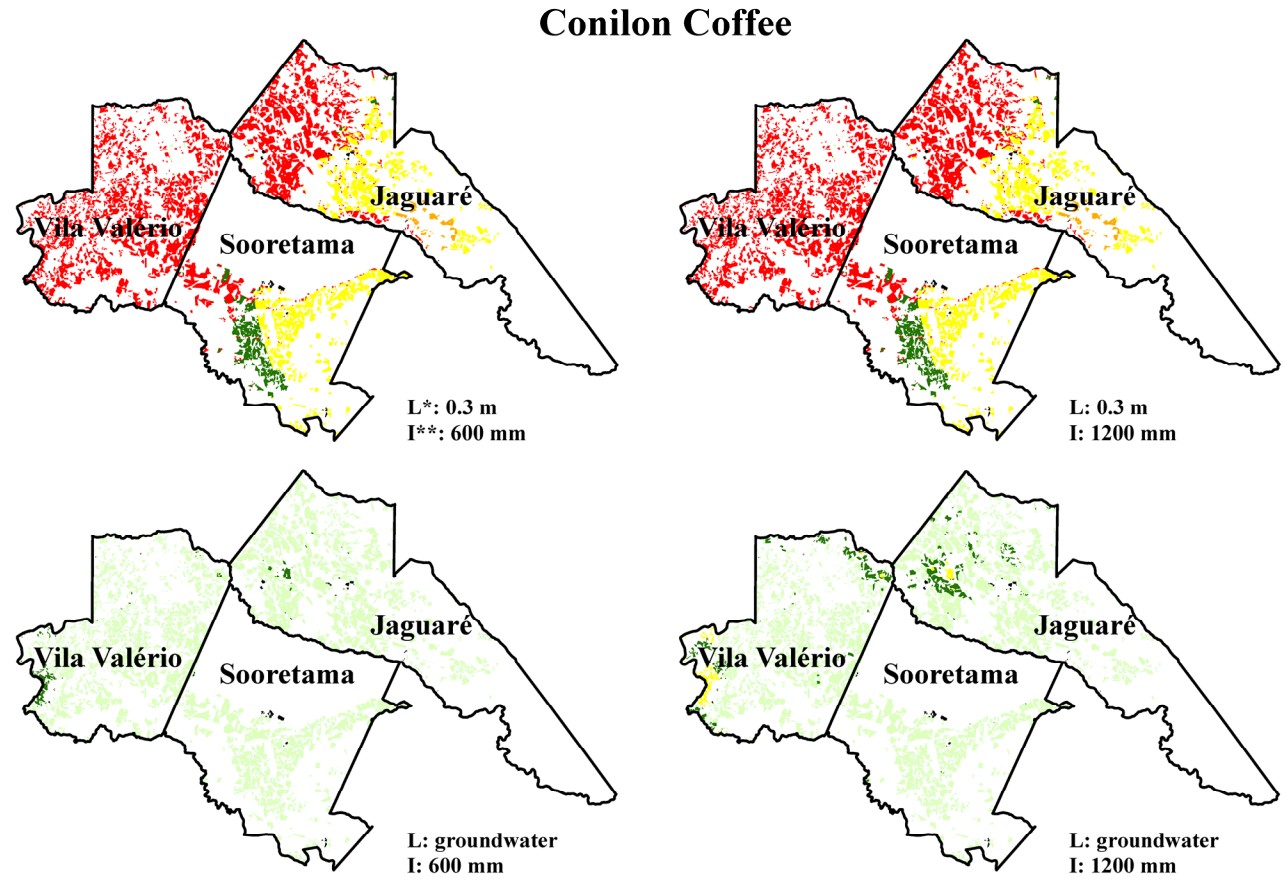

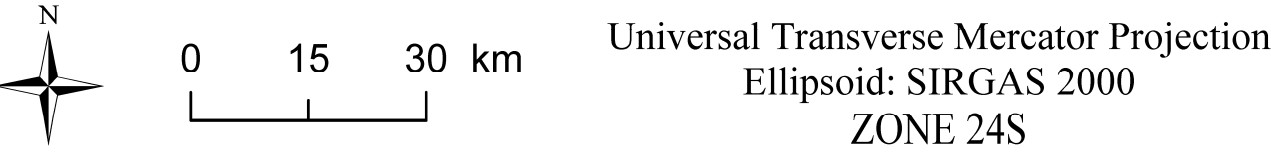

| Depth (m) | Legend | Potential leaching Sulfentrazone | Attenuation Factor (AF) | Jaguaré | | | | Vila Valério | | | | Sooretama | | | |
|---|---|---|---|---|---|---|---|---|---|---|---|---|---|---|---|
| | | | | 600 mm | | 1200 mm | | 600 mm | | 1200 mm | | 600 mm | | 1200 mm | |
| | | | | Area (km²) | % | Area (km²) | % | Area (km²) | % | Area (km²) | % | Area (km²) | % | Area (km²) | % |
| **0.3** | | Very low | 0.0 to 0.0001 | - | - | - | - | 0.64 | 0.56 | 0.64 | 0.56 | - | - | - | - |
| | | Low | 0.0001 to 0.01 | 1.03 | 0.84 | 1.03 | 0.84 | 0.03 | 0.02 | 0.03 | 0.02 | 19.66 | 25.76 | 19.66 | 25.76 |
| | | Medium | 0.01 to 0.1 | 42.81 | 35.04 | 42.81 | 35.04 | 0.02 | 0.02 | 0.02 | 0.02 | 39.86 | 52.22 | 39.86 | 52.22 |
| | | High | 0.01 to 0.25 | 4.75 | 3.89 | 4.75 | 3.89 | - | - | - | - | - | - | - | - |
| | | Very high | 0.25 to 1 | 72.94 | 59.71 | 72.94 | 59.71 | 112.45 | 99.24 | 112.45 | 99.24 | 15.94 | 20.88 | 15.94 | 20.88 |
| | | Null and Improper areas | - | 0.64 | 0.53 | 0.64 | 0.53 | 0.17 | 0.15 | 0.17 | 0.15 | 0.87 | 1.14 | 0.87 | 1.14 |
| | | **Total** | | **122.17** | **100.00** | **122.17** | **100.00** | **113.31** | **100.00** | **113.31** | **100.00** | **76.34** | **100.00** | **76.34** | **100.00** |
| **Groundwater level** | | Very low | 0.0 to 0.0001 | 120.01 | 98.23 | 109.23 | 89.40 | 109.53 | 96.67 | 99.33 | 87.66 | 74.59 | 97.71 | 75.33 | 98.68 |
| | | Low | 0.0001 to 0.01 | 1.52 | 1.25 | 10.95 | 8.96 | 3.57 | 3.15 | 9.67 | 8.53 | 0.87 | 1.14 | 0.13 | 0.18 |
| | | Medium | 0.01 to 0.1 | - | - | 1.36 | 1.11 | 0.03 | 0.03 | 4.12 | 3.64 | - | - | - | - |
| | | High | 0.01 to 0.25 | - | - | - | - | - | - | 0.02 | 0.01 | - | - | - | - |
| | | Very high | 0.25 to 1 | - | - | - | - | - | - | - | - | - | - | - | - |
| | | Null and Improper areas | - | 0.64 | 0.53 | 0.64 | 0.53 | 0.17 | 0.15 | 0.17 | 0.15 | 0.87 | 1.14 | 0.87 | 1.14 |
| | | **Total** | | **122.17** | **100.00** | **122.17** | **100.00** | **113.31** | **100.00** | **113.31** | **100.00** | **76.34** | **100.00** | **76.34** | **100.00** |

N

0     15     30  km

Universal Transverse Mercator Projection
Ellipsoid: SIRGAS 2000
ZONE 24S

**Figure 3.** Attenuation Factor (*AF*) of active ingredient sulfentrazone evaluated for areas cultivated with conilon coffee (*Coffea canephora* Pierre ex Froehner) for the municipalities of Jaguaré, Vila Valério and Sooretama, ES, Brazil. * L: Groundwater depth and ** I: Irrigation.

# Conilon Coffee

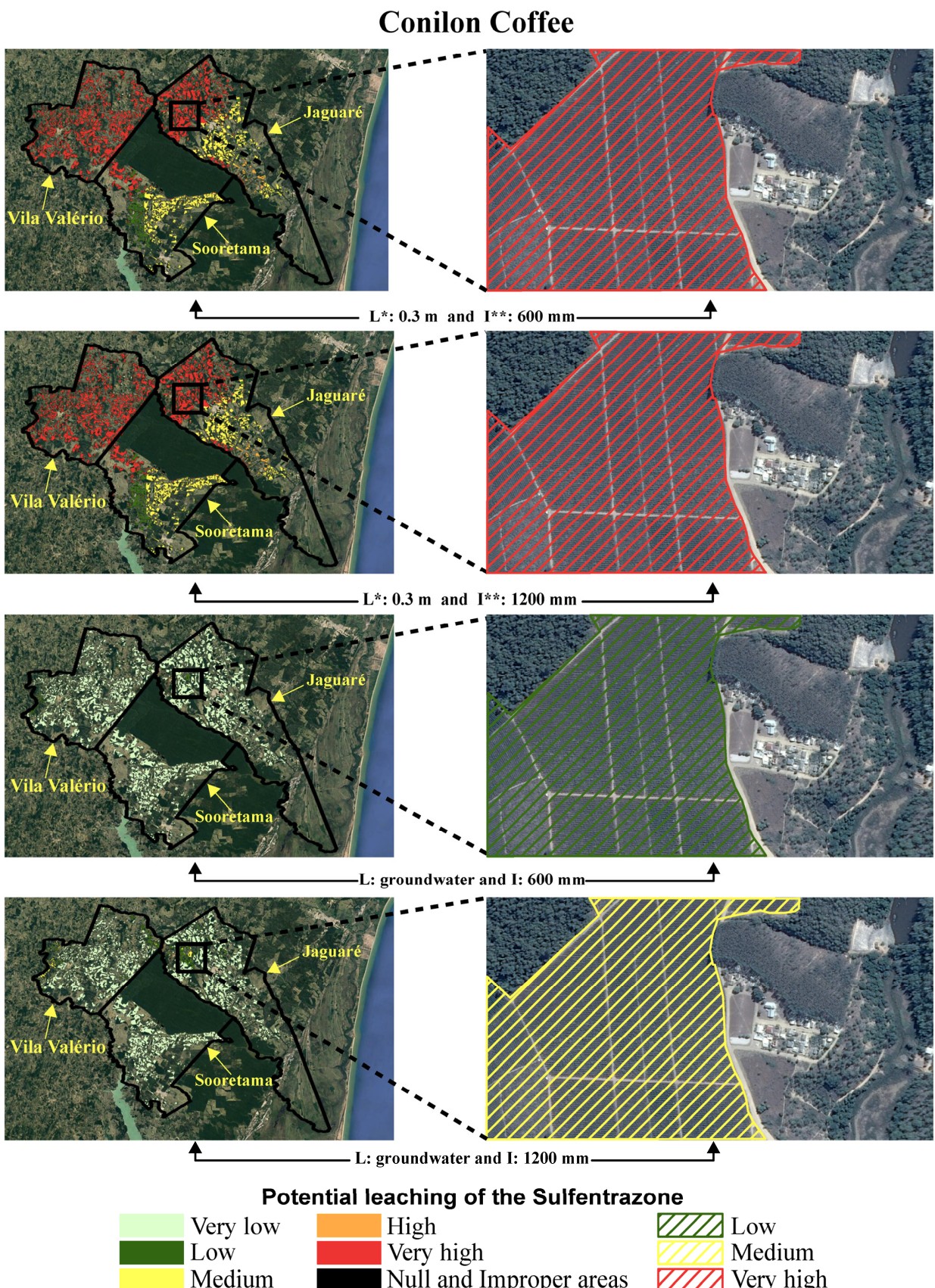

**Figure 4.** Attenuation Factor (*AF*) of active ingredient sulfentrazone evaluated for an extended study area cultivated with conilon coffee (*Coffea canephora* Pierre ex Froehner) for the municipality of Jaguaré, ES, Brazil. * L: Groundwater depth and ** I: Irrigation.

# Conilon Coffee

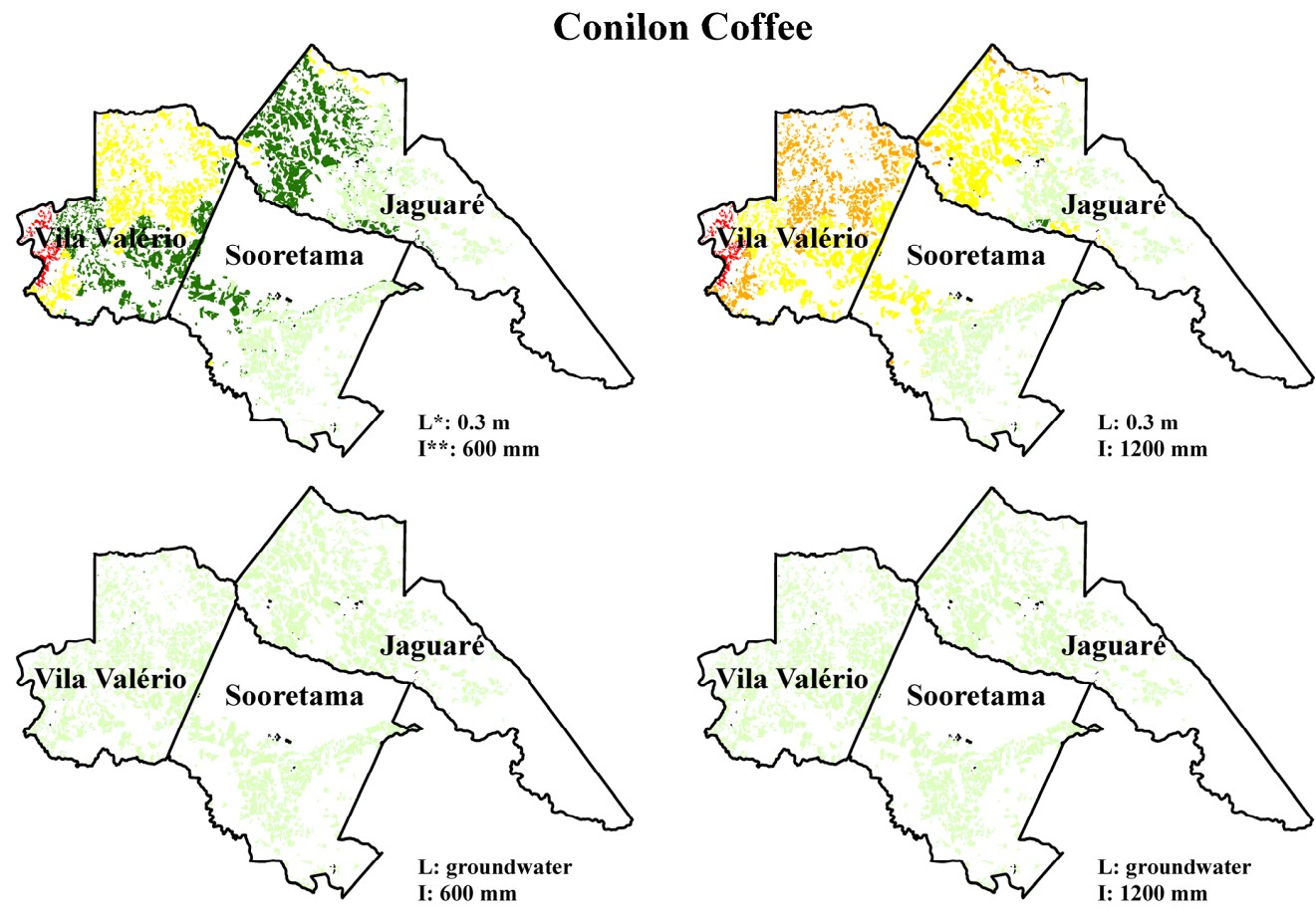

| Depth (m) | Legend | Potential leaching Thiamethoxam | Attenuation Factor (AF) | Jaguaré 600 mm Area (km²) | Jaguaré 600 mm % | Jaguaré 1200 mm Area (km²) | Jaguaré 1200 mm % | Vila Valério 600 mm Area (km²) | Vila Valério 600 mm % | Vila Valério 1200 mm Area (km²) | Vila Valério 1200 mm % | Sooretama 600 mm Area (km²) | Sooretama 600 mm % | Sooretama 1200 mm Area (km²) | Sooretama 1200 mm % |
|---|---|---|---|---|---|---|---|---|---|---|---|---|---|---|---|
| 0.3 | | Very low | 0.0 to 0.0001 | 48.60 | 39.78 | 48.60 | 39.78 | 0.78 | 0.69 | 0.78 | 0.69 | 59.52 | 77.97 | 59.52 | 77.97 |
| | | Low | 0.0001 to 0.01 | 68.82 | 56.33 | 0.82 | 0.67 | 50.02 | 44.15 | - | - | 15.72 | 20.59 | - | - |
| | | Medium | 0.01 to 0.1 | 4.11 | 3.36 | 68.00 | 55.66 | 56.26 | 49.65 | 50.02 | 44.15 | 0.23 | 0.30 | 15.72 | 20.59 |
| | | High | 0.01 to 0.25 | - | - | 4.11 | 3.36 | - | - | 56.26 | 49.65 | - | - | 0.23 | 0.30 |
| | | Very high | 0.25 to 1 | - | - | - | - | 6.07 | 5.36 | 6.07 | 5.36 | - | - | - | - |
| | | Null and Improper areas | - | 0.64 | 0.53 | 0.64 | 0.53 | 0.17 | 0.15 | 0.17 | 0.15 | 0.87 | 1.14 | 0.87 | 1.14 |
| | | **Total** | | **122.17** | **100.00** | **122.17** | **100.00** | **113.31** | **100.00** | **113.31** | **100.00** | **76.34** | **100.00** | **76.34** | **100.00** |
| Groundwater level | | Very low | 0.0 to 0.0001 | 121.53 | 99.48 | 121.53 | 99.48 | 113.13 | 99.84 | 113.13 | 99.84 | 75.46 | 98.85 | 75.46 | 98.85 |
| | | Low | 0.0001 to 0.01 | - | - | - | - | - | - | - | - | - | - | - | - |
| | | Medium | 0.01 to 0.1 | - | - | - | - | - | - | - | - | - | - | - | - |
| | | High | 0.01 to 0.25 | - | - | - | - | - | - | - | - | - | - | - | - |
| | | Very high | 0.25 to 1 | - | - | - | - | - | - | - | - | - | - | - | - |
| | | Null and Improper areas | - | 0.64 | 0.53 | 0.64 | 0.53 | 0.17 | 0.15 | 0.17 | 0.15 | 0.87 | 1.14 | 0.87 | 1.14 |
| | | **Total** | | **122.17** | **100.00** | **122.17** | **100.00** | **113.31** | **100.00** | **113.31** | **100.00** | **76.34** | **100.00** | **76.34** | **100.00** |

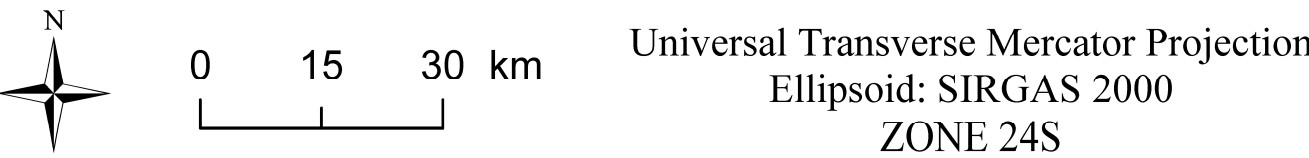

N

0    15    30 km

Universal Transverse Mercator Projection
Ellipsoid: SIRGAS 2000
ZONE 24S

**Figure 5.** Attenuation Factor (*AF*) of active ingredient thiamethoxam evaluated for areas cultivated with conilon coffee (*Coffea canephora* Pierre ex Froehner) for the municipalities of Jaguaré, Vila Valério and Sooretama, ES, Brazil. * L: Groundwater depth and ** I: Irrigation.

Figure 4 presents the spatial distribution of the Attenuation Factor (*AF*) of the active ingredient sulfentrazone evaluated for an expanded study area cultivated with conilon coffee in the municipality of Jaguaré, ES. Considering the root system depth of 0.30 m, for both irrigation levels (600 and 1200 mm), the study area presented very high PRL. However, at the groundwater depth, when simulating irrigations of 600 and 1200 mm, the study area presented low and medium PRL classes, respectively.

The soil class researches for the municipalities Jaguaré, Vila Valério and Sooretama, ES, revealed a predominance of argisol, gleysol and latosol soils. Given the *AF* indexes for the active ingredients sulfentrazone and thiamethoxam, in areas cultivated with conilon coffee, it was found that latosol is more favorable to leaching (Figure S17, Supplementary Material). This result is reinforced by the correspondence analysis (Figure S18, Supplementary Material), which showed heterogeneity between the percentage of occurrence of the *AF* index and soil types in the scenarios proposed for the municipalities of Jaguaré, Vila Valério and Sooretama ($p < 0.001$).

*3.5. Assessment of the Risk of Leaching in the Main Arabica Coffee Producing Municipalities*

The *AF* indexes for the sulfentrazone active ingredient evaluated for areas cultivated with arabica coffee in the municipalities of Brejetuba, Ibatiba and Iúna, ES, are presented in Figure 6 and Figure S19 of Supplementary Material. For areas cultivated with arabica coffee at the root system depth of 0.30 m, simulating irrigations of 600 and 1200 mm, the municipality of Ibatiba presented very high PRL equivalent to 99.25% (92.69 km$^2$), followed by the municipalities of Iúna (99.13% = 127.66 km$^2$) and Brejetuba (97.85% = 112.12 km$^2$). For the groundwater depth, for both irrigations (600 and 1200 mm), the three municipalities presented areas with mean PRL. However, the municipality of Iúna presented areas with high PRL (0.01% = 0.01 km$^2$) for both irrigation levels.

Figure 7 and Figure S20 of Supplementary Material show the evaluation of *AF* indexes of the active ingredient thiamethoxam for areas cultivated with arabica coffee in the municipalities of Brejetuba, Ibatiba and Iúna, ES. For areas cultivated with arabica coffee at a root system depth of 0.30 m, simulating irrigations of 600 and 1200 mm, the municipality of Ibatiba presented the very high PRL class in its territory (2.07% = 1.93 km$^2$), followed by the municipalities of Iúna (0.80% = 1.04 km$^2$) and Brejetuba (0.05% = 0.06 km$^2$). For the groundwater depth, for both irrigations, the three municipalities only presented areas with very low PRL.

The spatial distribution of the *AF* index of the active ingredient sulfentrazone evaluated for an enlarged study area cultivated with arabica coffee in the municipality of Iúna, ES, is presented in Figure 8. Considering the root system depth of 0.30 m, for both irrigations, the study area presented a very high PRL class. However, at the groundwater depth, for both irrigations, the study area showed, medium, low and very low PRL.

According to the soil class surveys for the municipalities of Brejetuba, Ibatiba and Iúna, argisol, cambisol, latosol and litholic neosol soils are predominant (Figure S20, Supplementary Material). In relation to the *AF* index, the litholic neosol soil stood out, presenting high potential risk of leaching, followed by the latosol, cambisol and argisol soils. However, the correspondence analysis (Figure S21, Supplementary Material) showed heterogeneity between the percentage of occurrence of the *AF* index and soil types in the proposed scenarios (depth and irrigation) for these municipalities ($p < 0.001$).

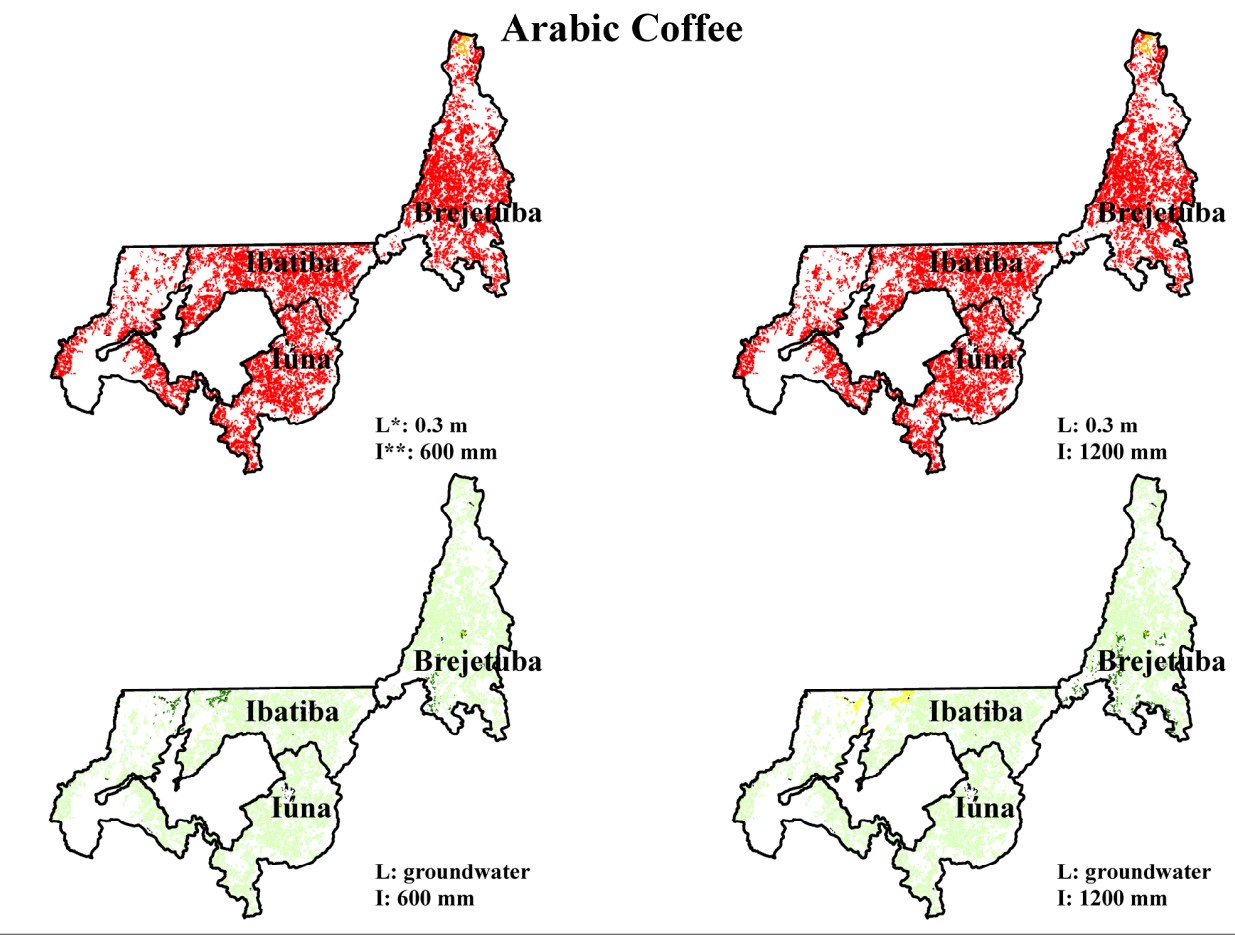

| Depth (m) | Legend | Potential leaching Sulfentrazone | Attenuation Factor (AF) | Brejetuba | | | | Ibatiba | | | | Iúna | | | |
|---|---|---|---|---|---|---|---|---|---|---|---|---|---|---|---|
| | | | | 600 mm | | 1200 mm | | 600 mm | | 1200 mm | | 600 mm | | 1200 mm | |
| | | | | Area (km²) | % | Area (km²) | % | Area (km²) | % | Area (km²) | % | Area (km²) | % | Area (km²) | % |
| **0.3** | | Very low | 0.0 to 0.0001 | - | - | - | - | - | - | - | - | - | - | - | - |
| | | Low | 0.0001 to 0.01 | - | - | - | - | - | - | - | - | - | - | - | - |
| | | Medium | 0.01 to 0.1 | - | - | - | - | - | - | - | - | - | - | - | - |
| | | High | 0.01 to 0.25 | 1.98 | 1.72 | 1.98 | 1.72 | - | - | - | - | - | - | - | - |
| | | Very high | 0.25 to 1 | 112.12 | 97.85 | 112.12 | 97.85 | 92.69 | 99.25 | 92.69 | 99.25 | 127.66 | 99.13 | 127.66 | 99.13 |
| | | Null and Improper areas | - | 0.49 | 0.43 | 0.49 | 0.43 | 0.69 | 0.74 | 0.69 | 0.74 | 1.12 | 0.87 | 1.12 | 0.87 |
| | | **Total** | | **114.59** | **100.00** | **114.59** | **100.00** | **93.39** | **100.00** | **93.39** | **100.00** | **128.78** | **100.00** | **128.78** | **100.00** |
| **Groundwater level** | | Very low | 0.0 to 0.0001 | 112.12 | 97.84 | 107.18 | 93.53 | 90.75 | 97.17 | 90.75 | 97.17 | 126.51 | 98.24 | 126.47 | 98.21 |
| | | Low | 0.0001 to 0.01 | 1.86 | 1.63 | 6.75 | 5.89 | 1.88 | 2.02 | 0.03 | 0.03 | 1.09 | 0.85 | 0.09 | 0.07 |
| | | Medium | 0.01 to 0.1 | 0.12 | 0.10 | 0.18 | 0.15 | 0.07 | 0.08 | 1.93 | 2.07 | 0.05 | 0.04 | 1.09 | 0.85 |
| | | High | 0.01 to 0.25 | - | - | - | - | - | - | - | - | 0.01 | 0.01 | 0.01 | 0.01 |
| | | Very high | 0.25 to 1 | - | - | - | - | - | - | - | - | - | - | - | - |
| | | Null and Improper areas | - | 0.49 | 0.43 | 0.49 | 0.43 | 0.69 | 0.74 | 0.69 | 0.74 | 1.12 | 0.87 | 1.12 | 0.87 |
| | | **Total** | | **114.59** | **100.00** | **114.59** | **100.00** | **93.39** | **100.00** | **93.39** | **100.00** | **128.78** | **100.00** | **128.78** | **100.00** |

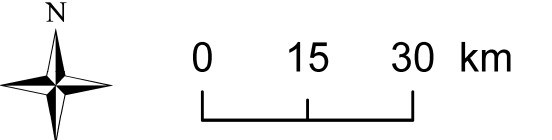

Universal Transverse Mercator Projection
Ellipsoid: SIRGAS 2000
ZONE 24S

**Figure 6.** Attenuation Factor (*AF*) of active ingredient sulfentrazone evaluated for areas cultivated with arabica coffee (*Coffea arabica* L.) for the municipalities of Brejetuba, Ibatiba and Iúna, ES, Brazil. * L: Groundwater depth and ** I: Irrigation.

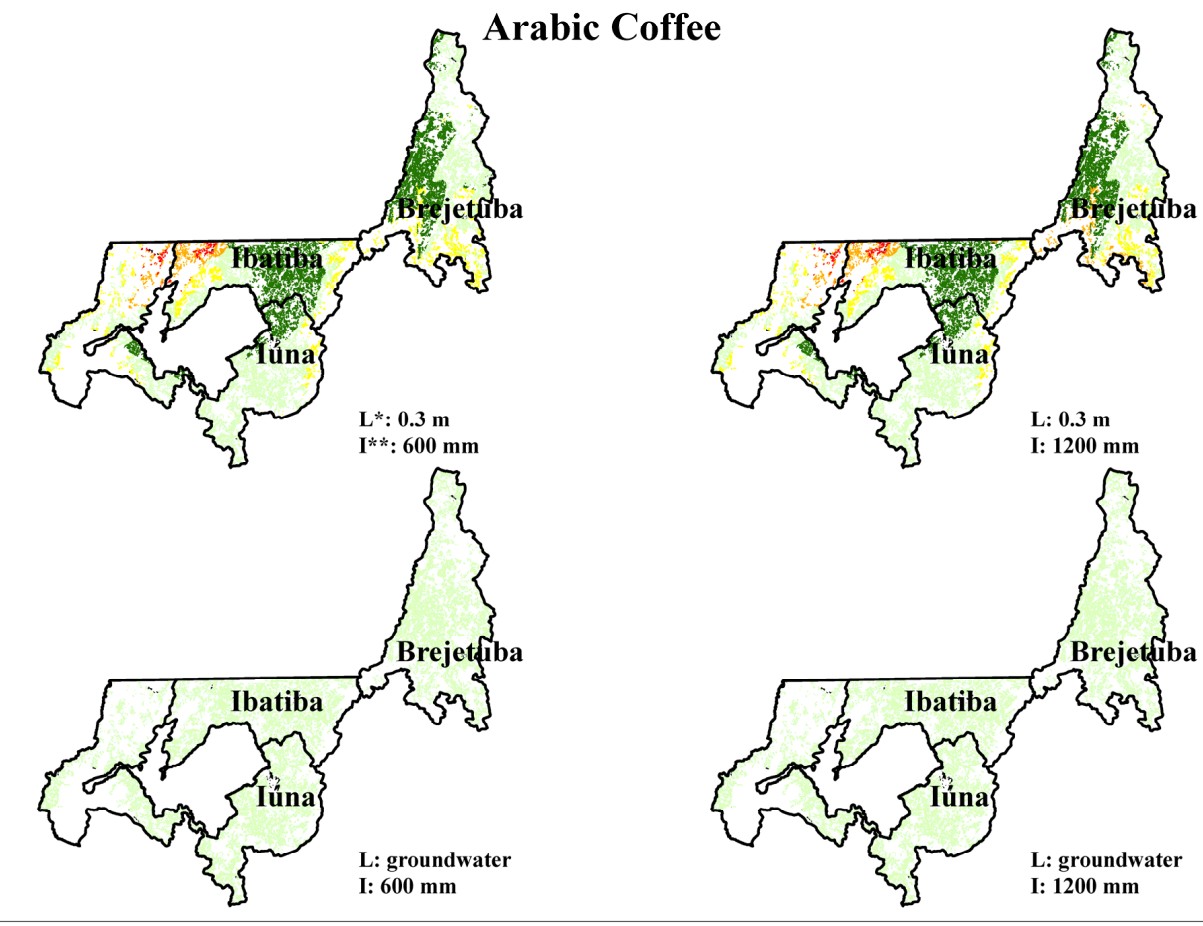

| Depth (m) | Legend | Potential leaching Thiamethoxam | Attenuation Factor (AF) | Brejetuba | | | | Ibatiba | | | | Iúna | | | |
|---|---|---|---|---|---|---|---|---|---|---|---|---|---|---|---|
| | | | | 600 mm | | 1200 mm | | 600 mm | | 1200 mm | | 600 mm | | 1200 mm | |
| | | | | Area (km²) | % | Area (km²) | % | Area (km²) | % | Area (km²) | % | Area (km²) | % | Area (km²) | % |
| 0.3 | | Very low | 0.0 to 0.0001 | 49.63 | 43.31 | 49.63 | 43.31 | 29.07 | 31.13 | 29.07 | 31.13 | 95.81 | 74.40 | 95.81 | 74.40 |
| | | Low | 0.0001 to 0.01 | 50.14 | 43.75 | 49.48 | 43.18 | 44.07 | 47.18 | 44.07 | 47.18 | 17.57 | 13.64 | 17.57 | 13.64 |
| | | Medium | 0.01 to 0.1 | 14.28 | 12.46 | 8.93 | 7.79 | 10.45 | 11.18 | 10.23 | 10.96 | 9.68 | 7.52 | 9.62 | 7.47 |
| | | High | 0.01 to 0.25 | - | - | 6.01 | 5.24 | 7.18 | 7.69 | 7.40 | 7.92 | 3.56 | 2.77 | 3.62 | 2.81 |
| | | Very high | 0.25 to 1 | 0.06 | 0.05 | 0.06 | 0.05 | 1.93 | 2.07 | 1.93 | 2.07 | 1.04 | 0.80 | 1.04 | 0.80 |
| | | Null and Improper areas | - | 0.49 | 0.43 | 0.49 | 0.43 | 0.69 | 0.74 | 0.69 | 0.74 | 1.12 | 0.87 | 1.12 | 0.87 |
| | | **Total** | | **114.59** | **100.00** | **114.59** | **100.00** | **93.39** | **100.00** | **93.39** | **100.00** | **128.78** | **100.00** | **128.78** | **100.00** |
| Groundwater level | | Very low | 0.0 to 0.0001 | 114.10 | 99.57 | 114.10 | 99.57 | 92.70 | 99.26 | 92.70 | 99.26 | 127.66 | 99.13 | 127.66 | 99.13 |
| | | Low | 0.0001 to 0.01 | - | - | - | - | - | - | - | - | - | - | - | - |
| | | Medium | 0.01 to 0.1 | - | - | - | - | - | - | - | - | - | - | - | - |
| | | High | 0.01 to 0.25 | - | - | - | - | - | - | - | - | - | - | - | - |
| | | Very high | 0.25 to 1 | - | - | - | - | - | - | - | - | - | - | - | - |
| | | Null and Improper areas | - | 0.49 | 0.43 | 0.49 | 0.43 | 0.69 | 0.74 | 0.69 | 0.74 | 1.12 | 0.87 | 1.12 | 0.87 |
| | | **Total** | | **114.59** | **100.00** | **114.59** | **100.00** | **93.39** | **100.00** | **93.39** | **100.00** | **128.78** | **100.00** | **128.78** | **100.00** |

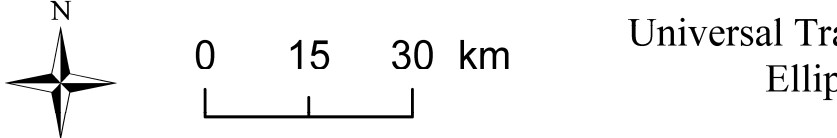

Universal Transverse Mercator Projection
Ellipsoid: SIRGAS 2000
ZONE 24S

**Figure 7.** Attenuation Factor (*AF*) of active ingredient thiamethoxam evaluated for areas cultivated with arabica coffee (*Coffea arabica* L.) for the municipalities of Brejetuba, Ibatiba and Iúna, ES, Brazil. * L: Groundwater depth and ** I: Irrigation.

# Arabic Coffee

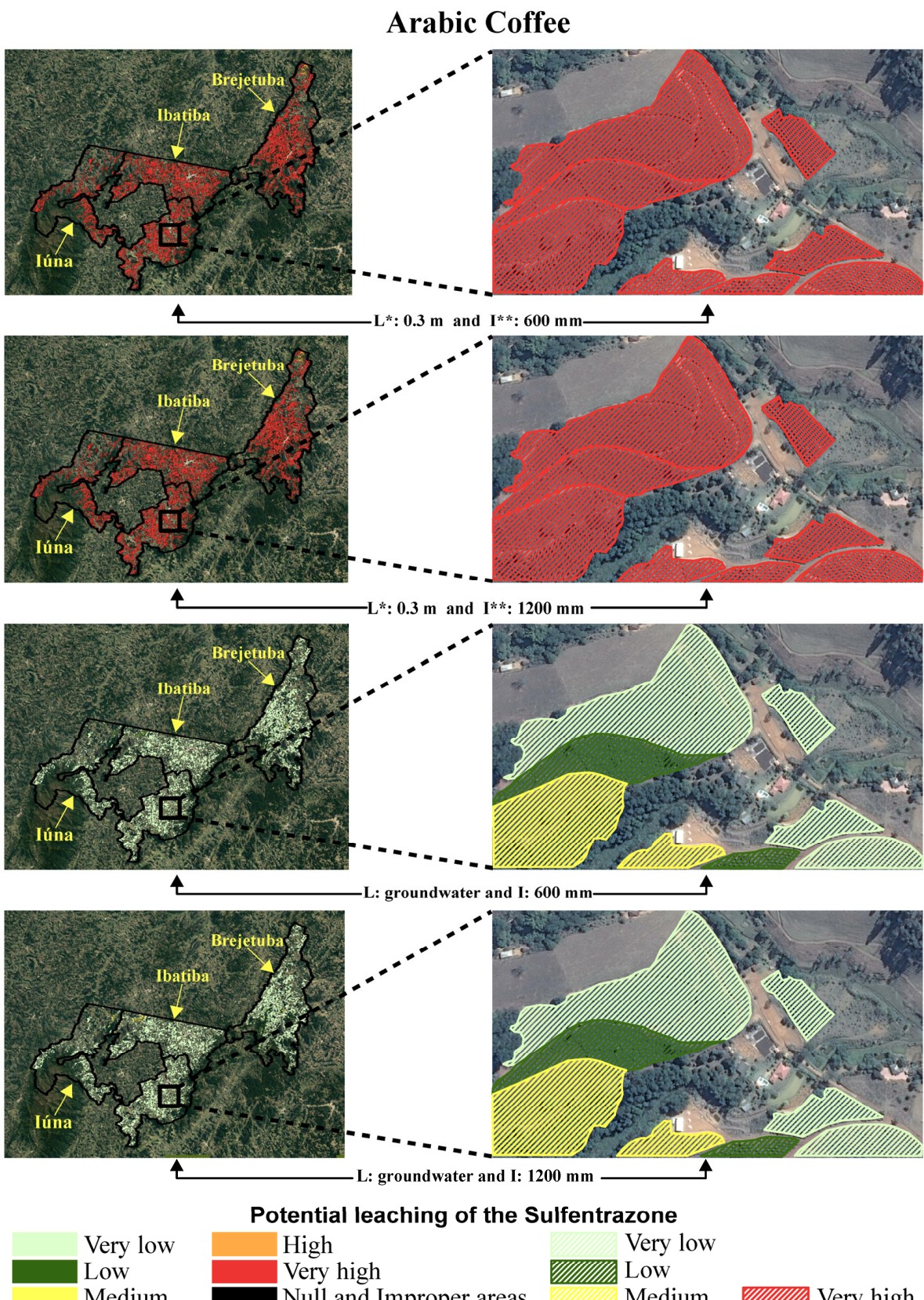

**Figure 8.** Attenuation Factor (*AF*) of active ingredient sulfentrazone evaluated for the extended study area cultivated with arabica coffee (*Coffea arabica* L.) for the municipality of Iúna, ES, Brazil. * L: Groundwater depth and ** I: Irrigation.

## 4. Discussion

Based on the results related to the edaphoclimatic zoning for conilon and arabica coffee cultivation in Espírito Santo state, Brazil—it was verified that the meteorological variables soil water deficit and temperature exert a strong influence on suitable, restricted and unsuitable areas. This was due to these variables being directly related to plant metabolic processes, such as the speed of the cellular chemical reactions that govern plant growth and photosynthetic performance, which depend on both water availability in the soil and the energy level measurements of the water-soil-plant-atmosphere system [40,56–58].

The areas being considered suitable for both conilon and arabica coffee are related to reliefs characterized by low altitudes in much of the state and high altitudes, mainly in the South and Southwest. Therefore, given the reduced average air temperature due to increasing altitude [40,56–58], the main municipalities producing arabica coffee are located in the elevated regions, while conilon coffee is produced in regions in Espirito Santo state, Brazil, with flat and gently undulating relief.

Regarding the soil classes in Espírito Santo, soils considered unsuitable (rock outcrop, spodosol, gleisol, quartzarenic neosol, indiscriminate soils) for development, for both conilon and arabica coffee, are mostly located in the lower areas near the Atlantic Ocean coastline. These soils present low chemical characteristics that prevent the full development of either crop.

The evaluated *LIX* and *GUS* methods allowed us to assess the probable leaching potential of the active ingredients studied in the water-soil-plant system using the physico-chemical characteristics. In the coffee production process, this preliminary analysis becomes a decisive factor for rational and efficient management to control and combat pests, diseases and weeds that generate production losses. The evaluated methods *LIX* and *GUS* do not simulate the transport of agrochemicals in a real field situation, but rather evaluate the potential leaching risk of an active ingredient allowing the comparison of an active ingredient with another one under the same environmental conditions [27,49]. Thus, according to the results, the classification of the ten active ingredients studied proved to be equivalent, showing that their physicochemical characteristics when used in different methods (*LIX* and *GUS*) presented the same results.

Among the ten active ingredients evaluated by the *LIX* and *GUS* methods, only sulfentrazone and thiamethoxam were classified with PRL. This is proven by the half-life ($t\frac{1}{2}$) and the adsorption coefficient of carbon ($K_{OC}$), which exert great influence on the downward flow of agrochemicals in the soil. Active ingredients that present greater mobility, such as sulfentrazone and thiamethoxam, with reduced $K_{OC}$ values (43 and 56.2, respectively), have a strong potential with regard to leaching. The higher the value, the greater the potential for contamination of groundwater and the longer the time required for breakdown during the downward flow of the agrochemical in the soil profile [27,49,59].

The results suggest that the use of a single method, which considers only the physico-chemical properties of agrochemicals (*LIX* and *GUS*), may not be sufficient to predict the potential risk of leaching of agrochemicals into groundwater. Therefore, the adoption of other methods, based not only on the physicochemical characteristics of the agrochemicals, but also on the soil characteristics and geoclimatic conditions of the study area, is justifiable, especially the *RF/AF* method used in this study.

According to the *RF* index for evaluated pesticides used in conilon and arabica coffee cultivation, the decisive independent variable for the adsorption of active ingredients into the soil was $K_{OC}$, followed by *OC* and other variables ($K_H$, $\rho$, $\delta$ and *FC*). Organic matter is cited as a major soil constituent responsible for the formation of possible chemical bonds with organic molecules, such as pesticides [49,60].

For the edaphoclimatically suitable zones for conilon and arabica coffee cultivation in Espírito Santo state, Brazil, a rational and efficient management of the water-soil-plant system is required when using the active ingredients sulfentrazone and thiamethoxam. These active ingredients present high mobility in sandy soils with low organic matter content and can reach deeper soil layers and consequently the groundwater. Such behavior

can be observed with fluvic neosol and quartzeneic soils for the conilon coffee culture and litholic neosol soils with arabica coffee. These results are similar to those reported by other authors [61–65] who evaluated the mobility of the active ingredients sulfentrazone and thiamethoxam in tropical soils, highlighting the need for care when using these molecules in the agroecosystem.

The integration of Geographic Information Systems (GIS) with environmental models of agrochemicals is currently an important technology, since it allows us to relate the leaching of agrochemicals to a set of environmental variables. Some examples of studies already realized can be found in [13,66].

Regarding the soil results, latosol soil showed higher leaching values when compared to argisol and gleisol soils. This is due to this soil being characterized as a deep soil with a stable, well drained granular structure that favors the vertical movement of solutes [50,67]. In general, they are also flat soils that are slightly undulated and rich in clay, but relatively poor in organic matter, with the exception of the former purple latosol. On the other hand, argisols are shallow soils, around 1 m, usually rich in clay and organic matter [67], which hinders the vertical movement of solutes and liquids through its profile. In relation to cambisol, argisol soil presents higher density and porosity values and lower field capacity and organic carbon values, factors that explain the reduced leaching potential (Table S1, Supplementary Material).

Although litholic neosol soil presents low *OC* values, high leaching values may be due to this soil class generally being sandy. Additionally, this soil presents rock fragments and gravel in its body or surface and is associated with undulated and strongly undulated reliefs, with steep slopes, characteristic of the mountainous region of Espírito Santo state that produces arabica coffee. Therefore, these results corroborate with those found by [68] who performed a physicochemical and mineralogical characterization of litholic neosol soils with materials of different origins in the region of Jaboticabal, SP, Brazil.

The latosol class is present both in conilon coffee producing municipalities (Jaguaré, Vila Valério and Sooretama) and in arabica coffee producing municipalities (Brejetuba, Ibatiba and Iúna). Therefore, its general characteristics such as good drainage and low organic matter favor the percolation of the active ingredients sulfentrazone and thiamethoxam, contributing to an increased risk of groundwater contamination in these cultivated areas.

In the steeper areas, common in the arabica coffee producing municipalities in Espírito Santo state, surface runoff (which has an inverse relationship to percolation) can contribute significantly to contamination of lakes and rivers, especially when these are close to plantations that require irrigation water, as well as the application of agrochemicals to control weeds, pests and diseases.

The climatic conditions of Espírito Santo state significantly affect the degree of pesticide leaching. Even knowing that high temperatures, characteristic of the months of November, December, January and February [69] may contribute to the further breakdown of active ingredients [50], it is necessary to increase our attention on pesticide applications precisely during these months, which constitute the rainy period in the State, since, rainfall favors pesticide leaching into the soil [70].

## 5. Conclusions

Among the ten active principles used for spatialization of the Attenuation Factor (AF), five (chlorpyrifos, glyphosate, paraquat, pendimethalin and terbuphos) showed very low leaching potential. The others (2,4-D, diuron, sulfentrazone, tebuconazole and thiamethoxam) indicated representative values for classes ranging from very low to very high leaching potential, with emphasis on sulfentrazone and thiamethoxam, which presented the highest leaching potentials at groundwater level. Thus, a rational and efficient management of the water-soil-plant system is required to prevent groundwater contamination problems, especially during the months of November, December, January and February for Espírito Santo state.

From a wider perspective of prevention, the protection of environmental resources from potentially dangerous chemicals, particularly the protection of groundwater from pesticides, must be included in a general strategy of land use management. In this sense, the study allowed us to evaluate the potential risk of agrochemical leaching in tropical soils cultivated with coffee using GIS techniques and the methodological proposal can be adapted for other agricultural areas and crops. The implemented methodology can be improved in future works through the insertion of new environmental and/or edaphoclimatic variables, aiming to better represent the real phenomena of the studied environments.

**Supplementary Materials:** The following supporting information can be downloaded at: https://www.mdpi.com/article/10.3390/w14091515/s1, Figure S1: Espírito Santo state, Brazil; Figure S2: Methodological steps necessary for the elaboration of the edaphoclimatic zoning for coffee conilon (*Coffea canephora* Pierre ex Froehner) and arabica (*Coffea arabica* L.) in Espírito Santo state, Brazil; Figure S3: Necessary methodological steps for spatialization and evaluation of leaching risk of agrochemicals active ingredients in areas of edaphoclimatic aptitude cultivated with conilon and arabica coffee in Espírito Santo state, Brazil, using the *RF/AF* method; Figure S4: Main municipalities producing conilon coffee in Espírito Santo state, Brazil; Figure S5: Main municipalities producing arabica coffee in Espírito Santo state, Brazil; Figure S6: Necessary methodological steps for spatialization and evaluation of leaching risk of agrochemicals active ingredients in areas cultivated with coffee for the main-producing municipalities in Espírito Santo state, Brazil, using the *RF/AF* method; Figure S7: Soil-climatic variables for Espírito Santo state, Brazil. (a) Average annual temperature (°C); (b) Annual water deficit (mm); and (c) Soil types; Figure S8: Aptitude trails for conilon coffee (*Coffea canephora* Pierre ex Froehner) for Espírito Santo state, Brazil. (a) Average annual temperature (°C); (b) Annual water deficit (mm); and (c) Soil types; Figure S9: Aptitude trails for conilon coffee (*Coffea arabica* L.) for Espírito Santo state, Brazil. a) Average annual temperature (°C); b) Annual water deficit (mm); and c) Soil types; Figure S10: Edafoclimatic zoning for coffee A) conilon (*Coffea canephora* Pierre ex Froehner) and B) arabica (*Coffea arabica* L.) in Espírito Santo state, Brazil; Figure S11: Edaphoclimatic aptitude classes for conilon (*Coffea canephora* Pierre ex Froehner) and arabica coffee (*Coffea arabica* L.) in Espírito Santo state, Brazil; Figure S12: Agrochemical Retardation Factor (*RF*) evaluated for the soil and climatic aptitude area of conilon coffee (*Coffea canephora* Pierre ex Froehner) in Espírito Santo state, Brazil; Figure S13: Agrochemical Retardation Factor (*RF*) evaluated for the soil and climatic aptitude area of arabica coffee (*Coffea arabica* L.) in Espírito Santo state, Brazil; Figure S14: *AF* index in relation to soil types of the main active ingredients with potential leaching risk for conilon (*Coffea canephora* Pierre ex Froehner) and arabica coffee (*Coffea arabica* L.) in Espírito Santo state, Brazil; Figure S15: Correspondence analysis between active ingredients Sulfentrazone and Thiamethoxam, in different depth and irrigation scenarios, and soils class, for conilon (*Coffea canephora* Pierre ex Froehner) and arabica coffee *(Coffea arabica* L.) in Espírito Santo state, Brazil; Figure S16: Attenuation Factor (*AF*) of active ingredients Sulfentrazone and Thiamethoxam evaluated for areas cultivated with conilon coffee (*Coffea canephora* Pierre ex Froehner) for the municipalities of Jaguaré, Vila Valério and Sooretama, ES, Brazil; Figure S17: Index *AF* in relation to soil types of active ingredients Sulfentrazone and Thiamethoxam evaluated for areas cultivated with conilon coffee (*Coffea canephora* Pierre ex Froehner) for the municipalities of Jaguaré, Vila Valério and Sooretama, ES, Brazil; Figure S18: Correspondence analysis of *AF* index of active ingredients Sulfentrazone and Thiamethoxam in different depth and irrigation scenarios and localities for conilon coffee (*Coffea canephora* Pierre ex Froehner) in Espírito Santo state, Brazil; Figure S19: Attenuation Factor (*AF*) of active ingredients Sulfentrazone and Thiamethoxam evaluated for areas cultivated with arabica coffee (*Coffea arabica* L.) for municipalities of Brejetuba, Ibatiba and Iúna, ES, Brazile; Figure S20: Index *AF* in relation to soil types of active ingredients Sulfentrazone and Thiamethoxam evaluated for areas cultivated with arabica coffee (*Coffea arabica* L.) for municipalities of Brejetuba, Ibatiba and Iúna, ES, Brazil; Figure S21: Correspondence analysis of *AF* index for active ingredients Sulfentrazone and Thiamethoxam in different depth and irrigation scenarios and localities for arabica coffee (*Coffea arabica* L.) in Espírito Santo state, Brazil; Table S1: Soils physicochemical properties; Table S2: Thermal Aptitude ranges for conilon (*Coffea canephora* Pierre ex Froehner) and arabica coffee (*Coffea arabica* L.); Table S3: Water aptitude ranges for the cultivation of conilon (*Coffea canephora* Pierre ex Froehner) and arabica coffee (*Coffea arabica* L.); Table S4: Adsorption potential categories for the Retardation Factor (*RF*).; Table S5: Leaching potential categories for the Attenuation Factor (*AF*) [71–79].

**Author Contributions:** Conceptualization, G.M.A.D.A.d.S., A.A.N. and A.R.d.S.; methodology, G.M.A.D.A.d.S., T.R.M. and A.R.d.S.; validation, G.M.A.D.A.d.S., T.R.M., F.R.P., P.A.G.F. and A.R.d.S.; formal analysis, G.M.A.D.A.d.S., V.T.d.Q., E.L.R., A.C.P.P., F.R.P. and M.H.d.S.; investigation, G.M.A.D.A.d.S., E.L.R., A.C.P.P., L.J.Q.T. and C.A.d.S.M.; resources, A.A.N., A.V.C., W.C.d.J.J. and A.R.d.S.; data curation, G.M.A.D.A.d.S., J.R.d.C., S.F.d.S., S.H.S., T.A.C. and P.A.G.F.; writing—original draft preparation, G.M.A.D.A.d.S., A.A.N., M.E.L.R.d.Q., R.S.J., C.A.d.S.M., M.H.d.S. and A.R.d.S.; writing—review and editing, G.M.A.D.A.d.S., V.T.d.Q., C.A.A.S.R., J.R.d.C., L.J.Q.T., S.H.S. and P.A.G.F.; visualization, G.M.A.D.A.d.S., C.A.A.S.R., S.F.d.S., R.S.J. and T.A.C.; supervision, A.A.N. and M.E.L.R.d.Q.; project administration, A.A.N. and A.R.d.S.; funding acquisition, A.A.N., A.V.C., W.C.d.J.J. and A.R.d.S. All authors have read and agreed to the published version of the manuscript.

**Funding:** This research received no external funding.

**Data Availability Statement:** The date presented in this study are available on request from the corresponding author.

**Acknowledgments:** The authors thank the promoting agency Coordination of Improvement of Higher Level Personnel (CAPES) and National Council for Scientific and Technological Development (CNPq), the government agencies National Institute of Meteorology (INMET), Brazilian Institute of Geography and Statistics (IBGE), Capixaba Institute for Research, Technical Assistance and Rural Extension of Espírito Santo (INCAPER) and Institute of Agricultural and Forestry Defense of Espírito Santo (IDAF) for providing part of the database. To the researchers that compose the research group of the CNPq Geotechnology Applied to Global Environment (GAGEN) for the help and dedication related throughout the research. Maps throughout this article were created using ArcGIS®software by Esri.

**Conflicts of Interest:** The authors declare no conflict of interest.

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
