# Peer review of "Potential Risk of Agrochemical Leaching in Areas of Edaphoclimatic Suitability for Coffee Cultivation"

_water, doi:10.3390/w14091515_

Round 1
Reviewer 1 Report
The manuscript “Potential risk of agrochemical leaching in areas of edaphoclimatic suitability for coffee cultivation”, by Gleissy Mary Amaral Dino Alves dos Santos and coworkers, presents interesting information on the implications of the use of agrochemicals in coffee crops, in particular, referred to the pollution of soil and water. Different methods were applied to assess the leaching potential of the most used agrochemical in a Brazilian state.
Concerning the structure of the manuscript, the authors tried to present the relevant background for their work in the introduction. However, in my opinion, there is theoretical information missing. I do not understand why the authors did not mention very important values close related to leaching problems such as the partition coefficient, Kd and the octanol/water distribution coefficient, Kow. Also, since the authors refer to the breakdown of agrochemicals, they must mention in the introduction the half-life of the agrochemicals. There are several references reporting ranges of values for the half-life for different pesticides in both water and soil. In fact, the authors use these values for the models.
In the Introduction, the authors should specify what "plant protection products" are. Are they pesticides? Are they a group of compounds that includes pesticides? It is unclear since the authors wrote "pesticides" and "plant protection products". Are they the same?
In the article, some sentences must be rephrased. I have included some corrections in the PDF document (attached) and suggested the revision of some specific sentences; the authors must perform a careful revision of the text and evaluate the use of certain terminology.
The authors should take into consideration that acronyms and non-conventional abbreviations require the presentation of their meaning the first time they are used.
The description of materials and methods is appropriate.
In the Supplementary Materials, figures must have labels. In Table S1, S4, I do not understand the meaning of the numbers (superscripts). The authors must clarify this aspect.
I don't understand why the authors have "Supplementary Materials 1" or "Supplementary Materials 2". It is senseless! Only one document is necessary (Supplementary Materials) and all tables and figures must be placed sorted in the same order as they are mentioned in the main document. My advice for the authors: Make it simple.
The references in the Supplementary Materials must be also included, in some way, in the list of references of the article. The references in the Supplementary Materials do not have the appropriate format. There are also mistakes concerning the numbers.
The conclusion is poor. I believe that the authors could improve this section even with the inclusion of perspectives and/or suggestions for future works.
In summary, I conclude that this paper requires some corrections. The Supplementary Materials also require a revision. Only after minor revisions, this paper can be considered for publication in Water.

Reviewer 2 Report
Dear authors,
The manuscript is an interesting study in the context of sustainable development, as it is necessary to ensure food for the entire population and to consider the water quality for food production in water stress conditions.
Specifically, my indications/ comments are:
- In this study, did you analyze the concentration of different pollutants or is it a simulation study?
- Please reconsider all the figure and table number since there is just one supplementary document (e.g. Lines: 84-85, 96, 99, etc).
- The figures numbers should be Supplementary materials Fig S1, S2, etc.). The notation should be consistently used.
- The RF should be used consistently. Is retarding factor (line 146) or is delay factor (line 175), Retardation Factor (line 65).
- The conclusion part should be improved by introducing a more pertinent conclusion of the work.
Thus, my decision is Minor revision.
Reviewer 3 Report
The manuscript is very well designed and written. The proposed study is very interesting and can be applied for other agriclutral areas and crops. However, there are some minor errors needs to be corrected.
-line 78: The emphasis on these two ingredients is part of the results, so its better to nominate the all ingredients here.
- The authores nominated Supplementary Materials 1 and 2, which means that there are two supplementary materials, but I found only one!
-line 132: Table 1 does not represents the physicochemical properties of the ingredients! Tables needs to be renumbered along the manuscript (check line 149-150).
- Figure's captions in the S.M. are missing
- Fig, 3, 4, 6 and 7 look blurry
